# Explaining the Mistakes of Neural Networks Using Latent Sympathetic Examples

## Abstract

Neural networks make mistakes. The reason why a mistake is made often remains a mystery. It would be useful to have a method that can give an explanation that is intuitive to a user as to why an image is misclassified. In this paper we develop a method for explaining the mistakes of an image classification model by visually showing what must be added to an image such that it is correctly classified. Our work combines the fields of adversarial examples, generative modeling and a correction technique based on difference target propagation to create an technique that creates explanations of why an image is misclassified. In this paper we explain our method and demonstrate it on MNIST and CelebA. This approach could aid in demystifying neural networks for a user.

## 1 Introduction

Given the increasingly widespread use of deep learning in real-world applications, it has become increasingly important to give explanations for specific images that are misclassfied.

For example, suppose a self-driving car, controlled by a neural network, makes a sudden decision to stop in the middle of the road, because it mis-identified a tree as a pedestrian who was about to cross. How could we understand that the reason for this decision was a poorly-trained pedestrian detector?

One approach we could take to"explain" these errors would be to use Gradient Descent learn a perturbation to the misclassified image such that it is correctly classified. We could then observe the perturbation and see what needed to be modified in the input to produce the correct class. In our example this might correspond to perturbing the car's camera image at the time of the stop, to minimize the strength of the "stop" output. We could then observe what change to the image was needed to avoid the sudden stop. We might observe that the tree which was mis-identified as a person becomes less person-like and conclude that a faulty pedestrian-detector was to blame.

This is the "Gradient Descent on the input" approach taken by (Szegedy et al., 2013), in which they introduced the notion of adversarial examples. Adversarial examples are created by perturbing an image so that it is misclassified. The surprising finding of this paper was that it is almost always possible to create an imperceptibly small perturbation to an image that changes the output class to any target value. These perturbations tend to look like white noise to the naked eye, and tell us nothing about what caused the image to be classified as it was.

Ideally, for the user to understand why a certain image is misclassified, the perturbations should be constrained to only the parts of an image relevant to the class and must be interpretable to the user.

In this paper, we generate explanations that align with human perception and are meaningful using generative models that perturb the features of a misclassified image such that it is correctly classified. The paper is structured in the following manner. In section 2 we introduce our proposed method. In section 3 the experiments are discussed. Finally, in section 4 we discuss the related work and in section 5 we will conclude.

## 2 METHOD

### 2.1 TERMINOLOGY

Our method (described visually in Figure 1), consists of a **classifier** ($c(\cdot)$) and a generative model, both trained separately on the same dataset. We refer to the discrete classes predicted by the classifier as $c(\cdot)$ and the class probabilities as $p_c(\cdot)$. We use the cross entropy error for our classifier denoted as $CE(\cdot, y)$. The generative model consists of an **encoder** ($e(\cdot)$) and a **generator** ($g(\cdot)$). We refer to the **original image** as $x$ and the correct class for the original image as y. Once the original image has been encoded to the latent space of the generative model ($e(x)$), we refer to it as $z$. The reconstruction of $z$ we call the **reconstructed image** and refer to as $g(z)$. Furthermore, we call $z + \epsilon$ the perturbation and $g(z+\epsilon)$ the **perturbed reconstruction**. Lastly, we call the result of our method $x'$ the **perturbed image**.

### 2.2 OUR METHOD

Formally, the problem we try to solve can be defined in the following manner. We have a misclassified image $\mathbf{x}$ (i.e.. $c(\mathbf{x}) \neq y$). We wish to generate $x'$, a perturbed version of $x$, such that $c(x') = y$. We define the perturbed reconstruction according to equation 1.

$$x' = g(z + \epsilon) + x - g(z) \tag{1}$$

We can see that if $\epsilon$ is 0, we are left with $x$. As such, so long as $g$ is smooth, the condition $\lim_{\epsilon \to 0} x' = x$ holds. This equation is inspired by Difference Target Propagation, proposed by (Lee et al., 2015), where they used it to propagate a perturbation backwards through a network. The goal of our method is to firstly, minimize the Cross Entropy: $CE(p_c(x'), y)$ and secondly to keep the euclidean norm of $\epsilon$ as small as possible to limit the amount of features added to the original image. We accomplish the first goal by using gradient descent to learn the perturbation $\epsilon$. As such epsilon is updated iteratively by formula 2. Where $\lambda$ is a small constant learning rate.

$$\epsilon_{n+1} = \epsilon_n - \lambda \frac{\partial CE(c(x'), y)}{\partial \epsilon_n} \tag{2}$$

The second goal is accomplished by stopping the perturbation of $\epsilon$ once the condition $c(x') = y$ is met. A visual representation of our method is shown in figure 1. Because we are minimizing the $CE(p_c(x'), y)$ using a generative model as opposed to maximizing it, we say that our method generates *Latent Sympathetic Examples*(LSE).

For this research we use a variational auto encoder (Kingma & Welling, 2013) as a generative model for MNIST and a combination of a variational auto encoder with a Generative Adversarial Network (Larsen et al., 2015) as a generative model for CelebA.

### 2.3 DEFINING WHAT IT MEANS TO EXPLAIN A CLASSIFICATION

Various definitions for explaining classifier decisions exist. We define an explanation for a classifier decision as; the perturbation that must be made to an image such that it is correctly classified. Subject to the conditions that:

1. The added parts are interpretable and preferably visible by the naked eye.

2. The perturbation is as close as possible to the original encoded point in the latent space of the generative model. This is measured as the euclidean norm of $\epsilon$.

Our method is encouraged to keep the euclidean norm of $\epsilon$ to a minimum. This is done such that method only adds those parts of the reconstruction that are most relevant for the original image to be perturbed into the correct class. This is relevant because otherwise the method could project a prototypical class example over the original image.

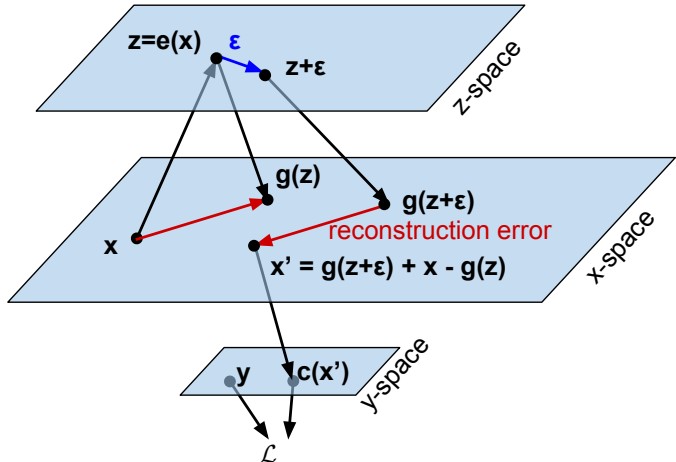

Figure 1: Latent Sympathetic Examples (LSE). The original input image is encoded and reconstructed . $x'$ is then perturbed, by iteratively calculating the gradients with respect to the correct class and updating $\epsilon$. By keeping $\epsilon$ as small as possible, much of the reconstruction error should be canceled out by the reconstructed image.

## 3 EXPERIMENTS

### 3.1 MNIST

We trained two different Variational Auto Encoders on MNIST. The first had a 2 dimensional latent space, the second a 10 dimensional latent space. We used the former in order to create a latent space that was easy to display visually, alongside the $CE(p_c(x'), y)$ errors. We used the latter to determine the impact of the dimensionality of the latent space on the rate of success of our method to push images into another class. We attached a classifier to the variational auto encoders and trained it without updating the weights of the VAE's. In order to get an intuition of how our method works, we discuss the results created using the 2-D VAE and their respective conclusions. Firstly, the predictions made by the classifier from the $g(z)$ may differ from the predictions made using $x'$. In order to illustrate this we have plotted the predictions $c(g(z))$ and the predictions of $c(x')$ for various values of $z$ and $\epsilon$, respectively. $c(g(z))$ is shown in figure 2 and $c(x')$ is shown in figure 3. As can be seen the predicted values are quite different. Secondly, our method may not be able to always perturb to the target class. Figure 3 shows that our method may not work for all cases. The class '1' is missing for this image. If this is the target class, then our method will not be able to generate a LSE for it. Furthermore, our method sometimes gets stuck in a local minimum. In that case it may be possible to create an LSE, however the method is unable to reach it. An example of this happening is shown in appendix B.

In order to illustrate how our our method works part of the latent space of a successfully perturbed image has been plotted in figure 4. This figure also demonstrates how the perturbation is visible in the image space. As can be seen the image quality is quite coarse. The VAE with the 10-D latent space yields qualitatively better results. Appendix A plots the perturbed image at each gradient step for a VAE with a 10-D latent space.

### 3.2 EFFECTS OF THE DIMENSIONALITY OF THE LATENT SPACE

Another goal of our research is to explore what the relation between the dimensionality of the latent space of the VAE is and effectiveness of our method to generate LSE's. We conduct 2 experiments in order to investigate this. Firstly, we perturb a set of misclassified examples and calculate the success rate. We define the success rate as what percentage of the time our method succeeds in finding a perturbation which pushes the perturbed image to the correct class. Secondly, we try to use our method to generate *latent adversarial examples* (LAE's), we take correctly classified images and try

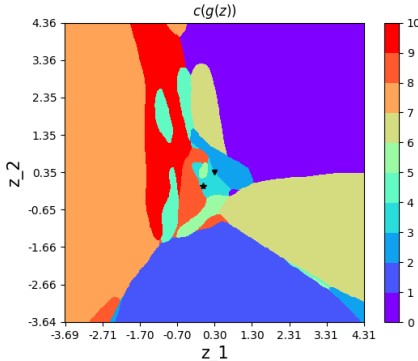 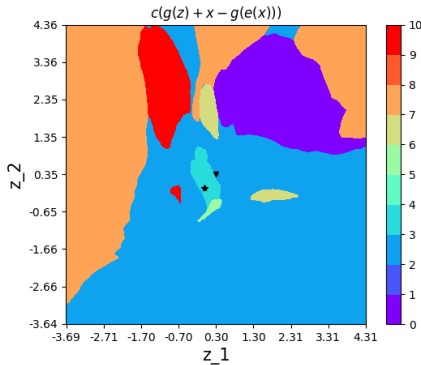

Figure 2: This figure shows the predictions of $c(g(z))$ given a grid of values for $z$. As can be seen here every MNIST class is present. The star labels the point where $z_1$ and $z_2$ is zero and the triangle where the original image is encoded.

Figure 3: The predictions of $c(x')$ given a grid of values for $\epsilon$. It can be seen that the classifier gives different results for the perturbed image $x'$ as opposed to the image reconstruction $(g(z)))$. Furthermore, not every class is available. Here it is impossible to perturb to a 1 for example. The star labels the point where $z_1$ and $z_2$ is zero and the triangle where the original image is encoded.

to perturb them to another class. We verify that our method should have more difficulty generating LAE's. However, it is only ineffective at generating LAE's using the 2D latent space, not the 10D latent space. Finally, we qualitatively review successfully generated LSE's when the VAE has a 2 and 10 dimensional latent space.

### 3.2.1 EFFECT OF THE DIMENSIONALITY OF THE LATENT SPACE OF THE VAE ON THE SUCCESS RATE OF FINDING A LSE

In order to explore what the impact of the dimensionality of the latent space is on the success rate of LSE generation, 2 experiments are run. We attempt to perturb 87 misclassified examples for both experiments to their correct class. We used a step size $\lambda$ of 0.1 . The method was successful in generating an LSE in 95.40% of the cases for the 10D VAE. The average euclidean distance from the original $e(x)$ points was 0.217. The 2D VAE successfully perturbed 78.16 % of the misclassified examples to their target class with an average euclidean distance of 0.10. This is evidence that the dimensionality of the latent space directly impacts the success rate of finding an LSE positively.

### 3.2.2 EFFECT OF THE DIMENSIONALITY OF THE LATENT SPACE OF THE VAE ON THE SUCCESS RATE OF FINDING AN LAE

In this experiment we want to check whether this method has more success in creating an LSE then creating a LAE. Therefore, an experiment is run where the goal is to perturb 100 correctly classified examples to a class different from their target class. Before generating the LAE's the images which already had the label that was being targeted were removed. As such each of the 10 resulting experiments (each of the MNIST labels was targeted), had a different number of images. The weighted average success rate of the 2-D and 10-D VAE's are 22.34 % and 91.11% respectively, the weighted average L-2 distances are 0.52 and 1.53, respectively. It can be seen that the average success rates for generating latent adversarial examples for the 2-D and 10-D VAE differ by quite a large margin.

An explanation for this difference is that it is harder to get stuck in a 10-D latent space as compared to a 2-D latent space, with their being more ways "down" in the error space. Furthermore, it can be seen that the LSE generation is more effective compared to the LAE generation. This may be due to the distance being traveled in the latent space. Because the target class in the LAE case is farther away then it is for LSEs on average, the chances of getting stuck in a local minimum are higher.

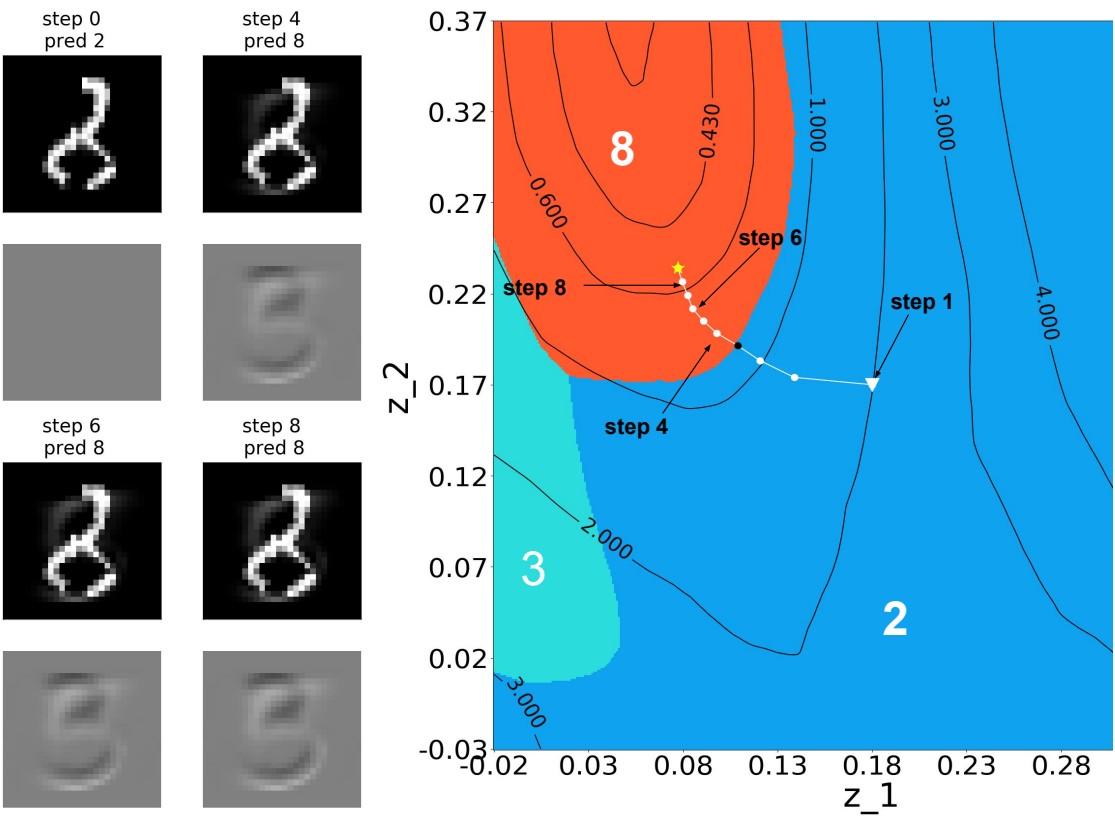

Figure 4: The plot on the right shows the path of $z + \epsilon$. The different colored areas show the class predictions made for a given value of $z + \epsilon$. These areas are annotated with the corresponding class. Furthermore, the contours indicate the cross entropy error $c(x', y)$ gives for different areas in the latent space. The triangle is the starting point, where $\epsilon$ is a vector of zeros. The black dot is where the perturbed image was first classified as the correct class, the gold star is where we stopped the algorithm. Several points are annotated with the correct step number. These steps correspond to images on the left. Here the LSE and the perturbation $(x' - x)$ are shown. As can be seen these perturbations are quite coarse.

### 3.2.3 QUALITATIVE IMPACT OF THE DIMENSIONALITY OF THE LATENT VARIABLE MODEL ON THE PERTURBED RECONSTRUCTION

In order for our method to succeed perturbations to the original image must firstly, be meaningful to a person and secondly be close to the original encoded point. The second condition is there to encourage that the perturbations added to the original image are limited to only those that are necessary. Other factors that can influence the size of the perturbations made in the image space are the quality of the reconstructions. A sample of perturbed images using a VAE with a latent space of 2-D and 10-D is shown in figure 5. These perturbations are visible by the naked eye. This image shows that the perturbations done by the latent variable model which used a 2-D latent space are much more coarse then its 10-D counterpart. Furthermore, the perturbations that were done using the latent variable model with the 10-D latent space are more precise. In the case of the four in the second row, the lower part of the long stroke on the left is dimmed a bit. The six in the third row has part of its circle dimmed and the top stroke is elongated and the eight in the final row that was misclassified as a zero was made less round by dimming some pixels in the middle and on the sides. The perturbations done by the model with the 2-D latent space sometimes are less precise. For example, the image in the top row has a six projected over it and the six in the third row is filled. These images are nonetheless, after this perturbation, correctly classified.

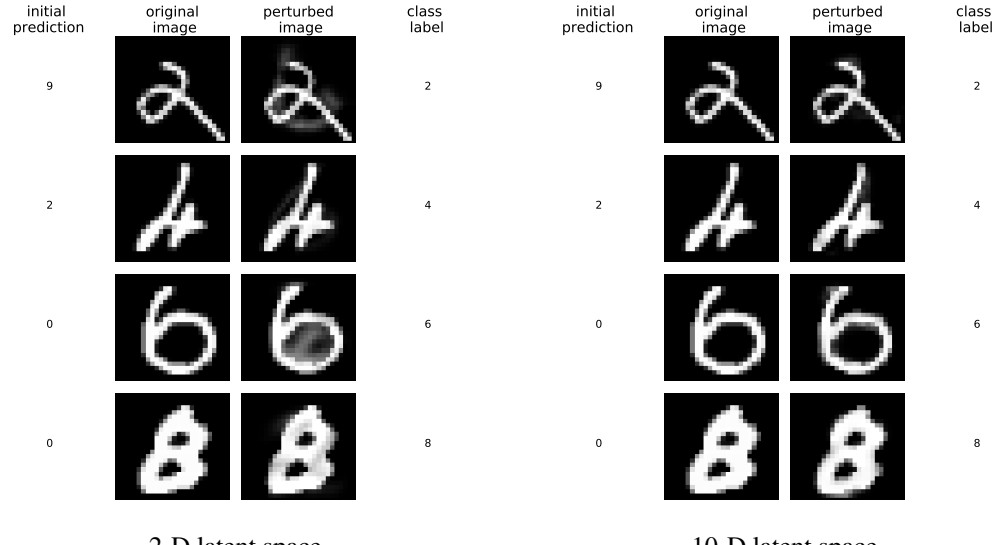

2-D latent space          10-D latent space

Figure 5: Comparison of the quality of the perturbed MNIST images and their latent dimensionality. From left to right for each image we have: 1) the original predicted class, 2) the original image, 3) the perturbed image and 4) the class label. All images were perturbed successfully. It can be seen from the images, that the latent variable model with the 2-D latent space has much coarser perturbations. In case of the top image it projected a large part of another image on top of it. However when we increase the latent space, the perturbations become much more subtle. Images best viewed digitally.

## 3.3 CELEBA

In order to test whether this method can be applied to get meaningful results on a more complicated dataset we applied this method on CelebA. We trained a VAEGAN with a 1024-D latent space on the dataset. We trained several classifiers on the traits; "Male", "Big Lips", "Receding Hairline", "Wearing Earrings", "Eyeglasses", "Wearing Necktie", "Mustache", "Sideburns", "Smiling" and "Rosy Cheeks". Several examples of successfully explained examples are shown in figure 6. It can be seen that the method gives plausible explanations as to why an image is misclassified. Some are visible only when looking at the perturbation $(x' - x)$.

Such explanations also have the potential to be useful in finding potential biases in the dataset. For example in image 10 in appendix B, images of males with long hair[1] are shown that were misclassified as female. By looking at a set of perturbations it becomes possible to find traits, which datapoints that are added to a dataset must have in order to remove bias. In this case that trait should be males with long hair.

We want to see whether the success depends on the trait. To this end we again calculated the success rates. These are shown in figure 1. As can be seen the success rate varies per trait, with the trait "Mustache" performing significantly worse compared to the other traits. We also show an image being perturbed to its correct class, this is shown in figure 8, appendix A.

| trait | Success Rate | trait | Success Rate |
|---|---|---|---|
| Male | 78 % | Receding Hairline | 95 % |
| Mustache | 15 % | Big Lips | 30 % |
| Rosy Cheeks | 21 % | Wearing Earrings | 98 % |
| Sideburns | 50 % | Eyeglasses | 86 % |
| Smiling | 78 % | Wearing Necktie | 94 % |

---

[1]Long Hair is not an annotated trait in the CelebA dataset. However cases such as these showed up in our experiment on the trait "Males" frequently.

Table 1: Results of the search for LSEs using the method on 100 misclassified examples of the CelebA dataset. As can be seen, the success of the method is dependent on the trait. The search is performed over the 1024D latent space of a VAEGAN trained on CelebA.

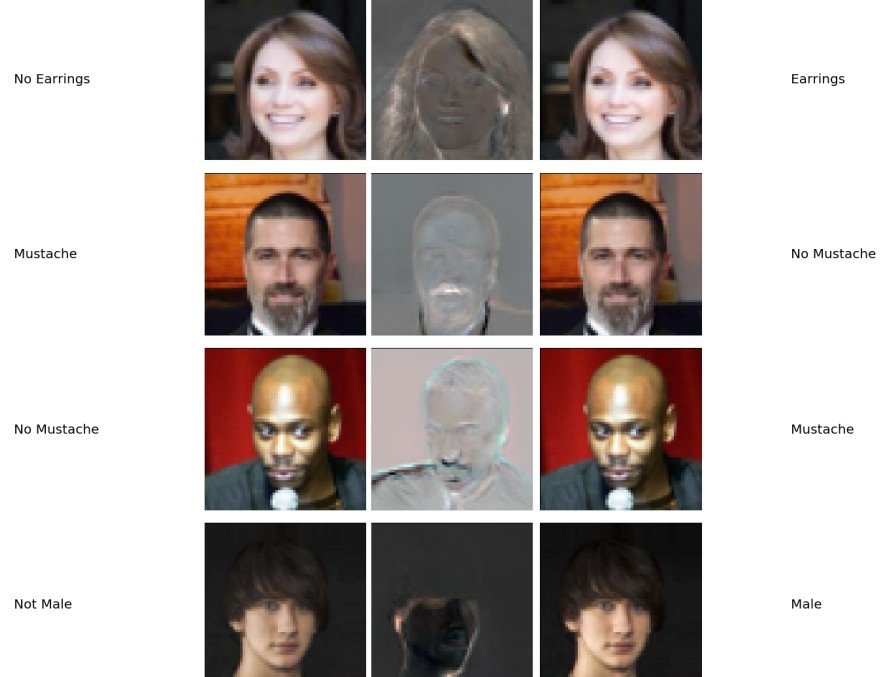

Figure 6: Examples of using the LSE technique to perturb images to be correctly classified. The left of center image is the original image, the center image is the perturbation ($x' - x$) and the right image is the perturbed image. The labels on the left denote the original (wrong) prediction and the labels on the right is the correct class. In the first row we see an image which is misclassified as not wearing earrings. By making the earring brighter the image is correctly classified. The second image from the top is of an image where the person was misclassified as having a mustache (the person has a beard), after making the hair below the nose lighter, the image was correctly classified as not having a mustache. The third image from the top was misclassified as not having a mustache. After darkening the hair under the nose, the image was correctly classified as having a mustache. The bottom image was misclassified as being female, after the perturbation made the adam's apple lighter the image was correctly classified. The perturbations are normalized using min-max normalization. The images are best viewed digitally.

## 4 RELATED WORK

The work done in our research is related to adversarial examples. Adversarial examples were introduced by (Szegedy et al., 2013). These examples take advantage of finding low probability pockets in the manifold on which the images are located, which fool the neural network into classifying the image in another class. As such an optimization problem can be formulated and solved to find adversarial examples. Such a problem can be optimized by means of various optimizers. For example, (Szegedy et al., 2013) used L-BFGS and (Goodfellow et al., 2014) took the sign of the gradients and used those to update the image until the prediction error was maximized. The prior two methods generate adversarial attacks in an untargeted fashion, meaning that they simply perturb to another unspecified class. Generating adversarial examples of a particular class is also possible. An example of this is the Deepfool algorithm (Moosavi-Dezfooli et al., 2016). More closely related to our work is the work done by (Kos et al., 2017). The authors devise several different adversarial attacks to change the encoding of an image in the latent space of a VAE or VAE-GAN such that the reconstructed image is different in a way that is advantageous to the attacker. Our work extends the prior work in the sense that it uses gradient descent to create an adversarial example and attempts to

create an explanation of why a certain classification was made . However there are two key differences. Firstly, as was said before, our method attempts to help the trained classifier to minimize the prediction error, in contrast to adversarial examples. As such it can be said such examples are not adversarial at all but sympathetic to the classifier. Secondly, despite one of the goals of adversarial attacks being to better understand the input-output mapping of a neural network, classical adversarial examples are non-intuitive to any user. As such our method constrains the search for adversarial examples by means of a trained generative model. The intuition being that the search is constrained to features that should be interpretable to a user.

Our work was in part inspired by the Difference Target Propagation technique developed by (Lee et al., 2015). They propose an alternative to backpropagation for assigning credit in deep networks, called "target propagation". During training, target propagation sends targets, instead of gradients, backwards through the network, where the target is a small perturbation on the activation of the layer from the forward pass. The authors show that so long as layers are invertible, this will minimize the final loss. Because layers are in general not invertible, they apply a linear correction (which we borrow in equation 1) to adjust shift the backpropagating target so that it remains close to the forward activation value. We use the same correction in our paper to ensure that as our latent perturbation becomes small, so does our perturbation in image space. i.e. $\lim_{\epsilon \to 0} x' = x$.

The work in this paper is also related to heatmapping methods. Heatmapping methods indicate which pixels contributed to a specific classification decision. Specifically, it is related to the image specific salieny map technique introduced by Simonyan et al. (2013). A downside of this technique is that the heatmaps are noisy due to them providing local explanations (Samek et al., 2017). This means that the irrelevant background parts of an image can also be changed in order to increase the classification score for the target class. For example, an image of a tree on an empty field of grass can be made more "tree" like for the classifier by putting tree like structures on the empty field. Our method often produces more localized perturbations.

## 5 DISCUSSION

In this paper we introduced a new method to construct an explanation for why a neural network has misclassified an image that is intuitive for a user to understand. We have demonstrated that the method gives meaningful results on both simple and more complicated datasets. Though the method looks promising, there is still a potential problem with applying it in the real world.

A problem may be that the method does not work well on all annotated traits that images may have. We already saw in the experiments on CelebA that the success rate depends on the trait. An explanation for this may be that the opinion of when an image has a certain trait may be ambiguous for annotators. For example, what constitutes somebody having big lips may vary among individuals to such an extent that only the extremes can be classified with certainty, making them hard to reach using gradient descent.

An application for our method, is to find class imbalances in data sets. An example was already given in the paper, however another method may also be possible. By averaging the difference between the perturbed reconstruction and the reconstructed image on a set of misclassified images. A heatmap can be constructed to find which area is most often perturbed. This area can point to a potential bias in the dataset.

Another application for our method is improving classifier performance. We can use our method to generate edge cases, by finding a class boundary and jumping in and out of it on various points. These edge cases can then be annotated by a human and used for training. A hurdle to widespread use of neural networks is that they can seem like black-boxes, when they feel it is difficult to understand why. We believe that this method will help overcome this problem by making the decisions explainable.

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

## A    PERTURBATION PROCESS EXAMPLES

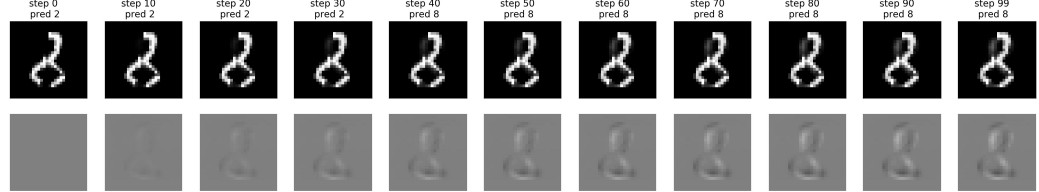

Figure 7: This plot shows the results of perturbing a misclassified image after different amounts of iterations. As can be seen the perturbation is qualitatively much more fine grained when compared to the perturbation in figure 4.

## B    LOCAL MINIMUM

## C    APPLICATION FOR DETECTING FEATURE IMBALANCES IN CLASSES

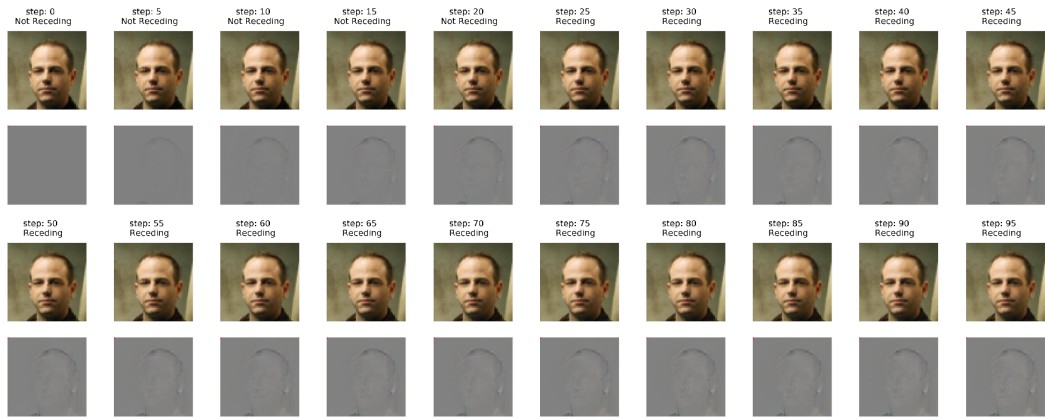

Figure 8: Plotting every fifth step through the latent space, alongside the perturbation $(x' - x)$. It can be seen that our method perturbs the original image to be correctly classified. The perturbations are normalized using min-max normalization, where the minimum is set to -1 and the maximum is set to 1.

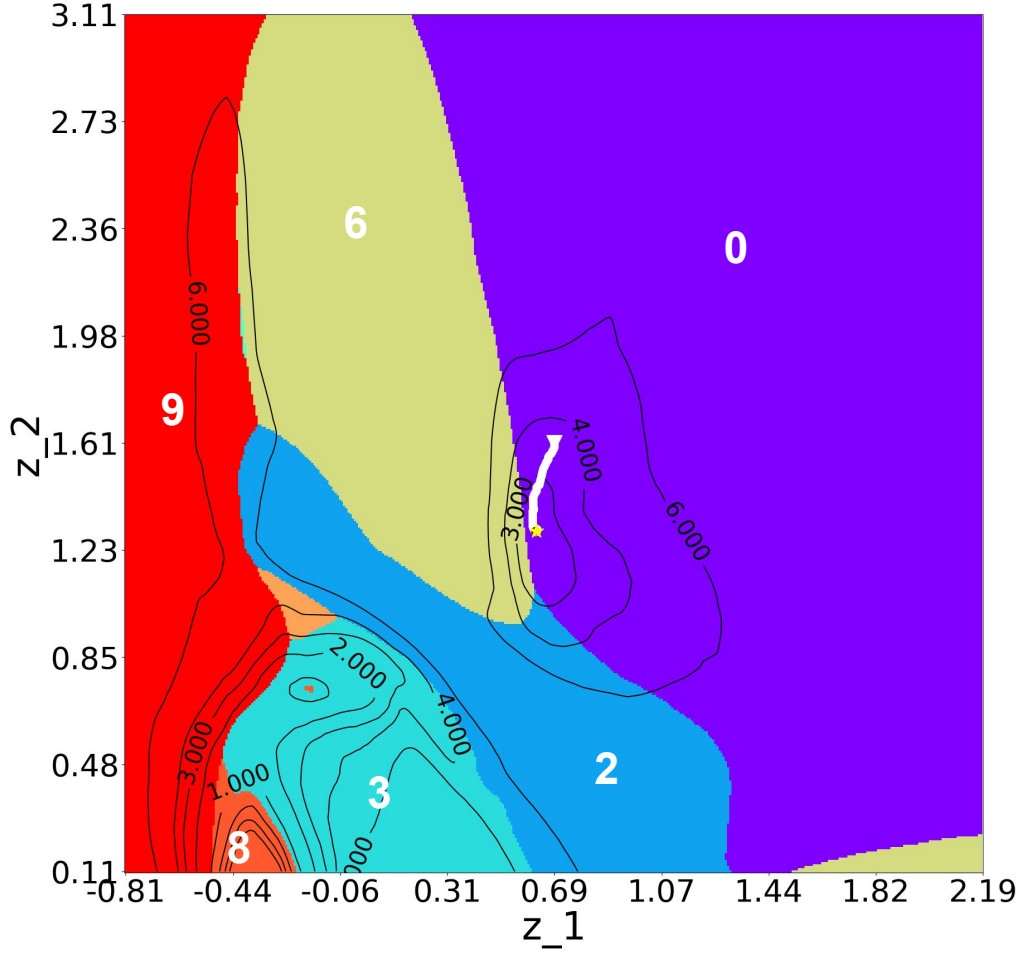

Figure 9: This plot shows our method getting stuck in a local minimum. The image was originally misclassified as a zero. The correct class, eight, exists, however a local minimum blocks the way. The white triangle is the starting point, the gold star is the end point of the search.

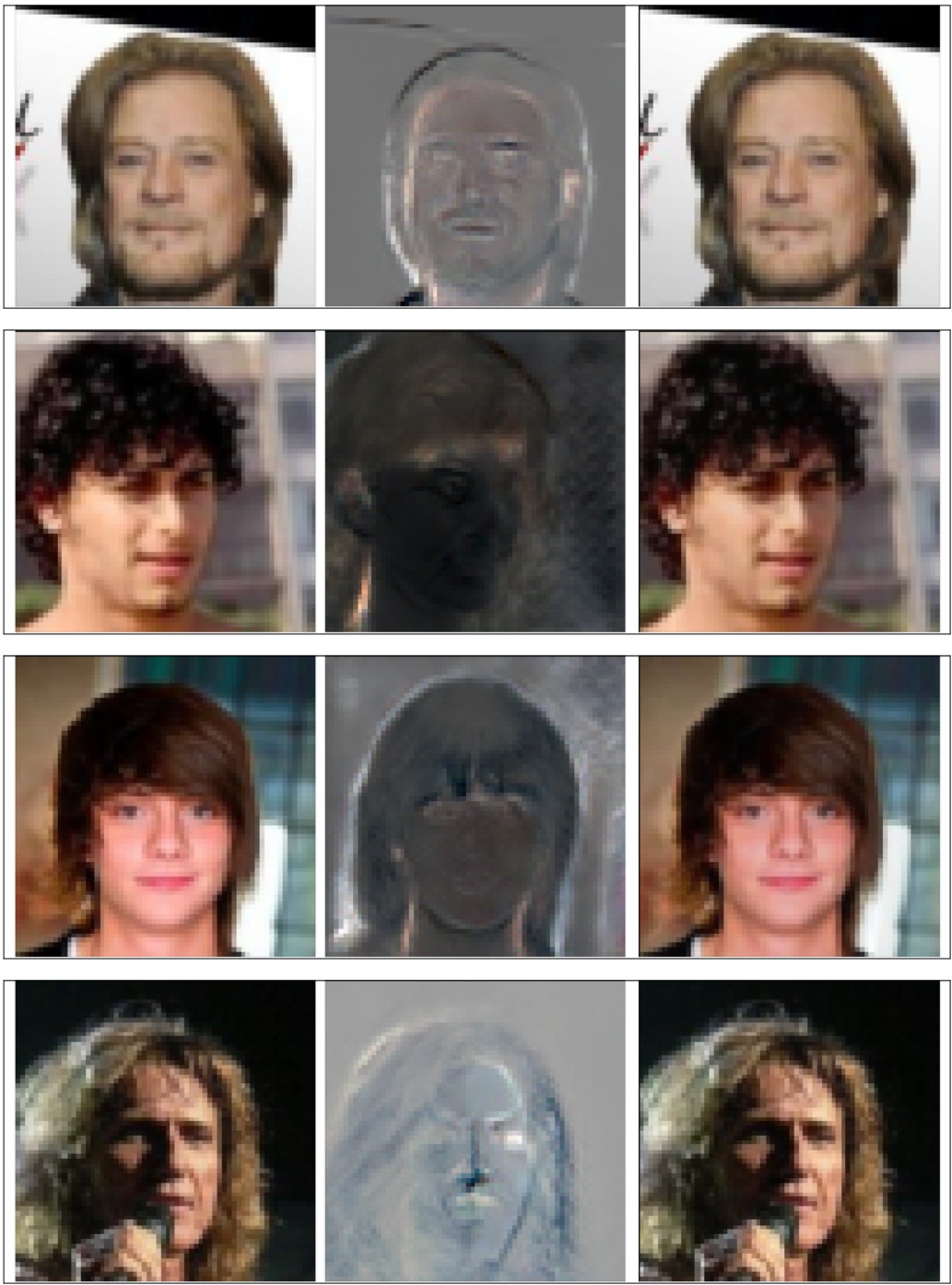

Figure 10: Image showing various men with long hair which were misclassified as "not male". Original image is shown on the left, the perturbation ($\mathbf{x}'$ - $\mathbf{x}$) is shown in the middle and the perturbed image is shown on the right. It can be seen that a lot of the perturbations target the hair. This may indicate insufficient men with long hair in the dataset, causing a bias. The perturbations are normalized using min-max normalization.

