# OpenReview forum: "Explaining the Mistakes of Neural Networks with Latent Sympathetic Examples"
_ICLR.cc/2018/Conference — Reject_

### Official Review · AnonReviewer1 · 2017-11-27
**Limited technical novelty. Unclear applicational benefit.**

**Rating:** 4
**Confidence:** 5

**Review:**

This paper proposes a method for explaining the classification mistakes of neural networks. For a misclassified image, gradient descent is used to find the minimal change to the input image so that it will be correctly classified.

My understanding is that the proposed method does not explain why a classifier makes mistakes. Instead, it is about: what can be added/removed from the input image so that it can be correctly classified. Strictly speaking, "Explaining the Decisions of Neural Networks" is not the most relevant title for the proposed method.

Based on my understanding about what the paper proposes, I am not sure how useful this method is, from the application point of view. It is unclear to me how this method can shed light to the mistakes of a classifier.

The technical aspects of the paper are straight forward optimization problem, with a sensible formulation and gradient descent optimization problem. There is nothing extraordinary about the proposed technique.

The method assumes the availability of a generative model, VAE. The implicit assumption is that this VAE performs well, and it raises a concern about the application domains where VAE does not work well. In this case, would the visualization reflect the shortcoming of VAE or the mistake of the classifier?

---

> ### Author Response · Authors · 2018-01-05
> **Reply at the top!**
>
> Hello AnonReviewer1,
>
> Just a notification that we have given our reply at the top of the page! In one comment for all reviewers.
>
> This is done to keep this forum tidy!
>
> Kind regards

---

### Official Review · AnonReviewer3 · 2017-11-28

**Rating:** 4
**Confidence:** 3

**Review:**

In this paper, the authors aim to better understand the classification of neural networks. The authors explore the latent space of a variational auto encoder and consider the perturbations of the latent space in order to obtain the correct classification. They evaluate their method on CelebA and MNIST datasets.

Pros:

 1) The paper explores an alternate methodology that uses perturbation in latent spaces to better understand neural networks
2) It takes inspiration from adversarial examples and uses the explicit classifier loss to better perturb the $z$ in the latent space
3) The method is quite simple and captures the essence of the problem well

Cons:
The main drawback of the paper is it claims to understand working of neural networks, however, actually what the authors end up doing are perturbations of the encoded latent space. This would evidently not explain why a deep network generates misclassifications for instance understanding the failure modes of ResNet or DenseNet cannot be obtained through this method. Other drawbacks include:

1)     They do not show how their method would perform against standard adversarial attack techniques, since by explaining a neural network they should be able to guard against attacks, or at-least explain why they work well.
2) The paper reports results on 2 datasets, out of which on 1 of them it does not perform well and gets stuck in a local minima therefore implying that it is not able to capture the diversity in the data well.

3) The authors provide limited evaluation on a few attributes of CelebA. Extensive evaluation that would show on a larger scale with more attributes is not performed.

4) The authors also have claimed that the added parts should be interpretable and visible. However, the perturbations of the latent space would yield small $\epsilon$ variation in the image and it need not actually explain why the modification is yielding a correct classification, the same way an imperceptible adversarial attack yields a misclassification. Therefore there is no guarantee that the added parts would be interpretable. What would be more reasonable to claim would be that the latent transformations that yield correct classifications are projected into the original image space. Some of these yield interpretations that are semantically meaningful and some of these do not yield semantically meaningful interpretations.

5)  Solving mis-classification does not seem to equate with explaining the neural network, but rather only suggest where it makes mistakes. That is not equal to an explanation about how it is making a classification decision. That would rather be done by using the same input and perturbing the weights of the classifier network.

In conclusion, the paper in its current form provides a direction in terms of using latent space exploration to understand classification errors and corrections to them in terms of perturbations of the latent space. However, these are not conclusive yet and actually verifying this would need a more thorough evaluation.

---

> ### Author Response · Authors · 2018-01-05
> **Reply at the top!**
>
> Hello AnonReviewer3,
>
> Just a notification that we have given our reply at the top of the page! In one comment for all reviewers.
>
> This is done to keep this forum tidy!
>
> Kind regards

---

### Official Review · AnonReviewer2 · 2017-12-03
**Interesting explanation for images, but lacks discussion about perturbations in the latent space which is most interesting part.**

**Rating:** 6
**Confidence:** 4

**Review:**

Summary: The authors propose a method for explaining why neural networks make mistakes by learning how to modify an image on a mistaken classification to make it a correct classification. They do this by perturbing the image in an encoded latent space and then reconstructing the perturbed image.  The explanation is the difference between the reconstructed perturbed encoded image and the reconstructed original encoded image.

The title is too general as this paper only offers an explanation in the area of image classification, which by itself, is still interesting.

A method for explaining the results of neural networks is still open ended and visually to the human eye, this paper does offer an explanation of why the 8 is misclassified. However, if this works very well for MNIST, more examples should be given. This single example is interesting but not sufficient to illustrate the success of this method.

The examples from CelebA are interesting but inconclusive. For example, why should adding blue to the glasses fix the misclassification. If the explanation is simply visual for a human, then this explanation does not suffice. And the two examples with one receding and the other not receding hairlines look like their correct classifications could be flipped.

Regarding epsilon, it is unclear what a small euclidean distance for epsilon is without more examples. It would also help to see how the euclidean distance changes along the path.  But also it is not clear why we care about the size of epsilon, but rather the size of the perturbation that must be made to the original image, which is what is defined in the paper as the explanation.

Since it is the encoded image that is perturbed, and this is what causes the perturbations to be selective to particular features of the image, an analysis of what features in the encoded space that are modified would greatly help in the interpretability of this explanation. The fact that perturbations are made in the latent space, and that this perturbation gets reflected in particular areas in the reconstructed image, is the most interesting part of this work.  More discussion around this would greatly enhance the paper, especially since the technical tools of this method are not very strong.

Pros: Interesting explanation, visually selects certain parts of the image relevant to classification rather than obscure pixels

Cons: No discussion or analysis about the latent space where perturbations occur. Only one easy example from MNIST shown and examples on CelebA are not great. No way (suggested) to use this tool outside of image recognition.

---

> ### Author Response · Authors · 2018-01-05
> **Reply at the top!**
>
> Hello AnonReviewer2,
>
> Just a notification that we have given our reply at the top of the page! In one comment for all reviewers.
>
> This is done to keep this forum tidy!
>
> Kind regards

---

### Author Response · Authors · 2018-01-05
**Reply to our reviewers**

Dear Anonymous Reviewers,

Thank you for taking the time to respond to our paper. We have taken your points for improvement into consideration and made  changes to the paper.

What we have done is the following. Firstly, we have changed the title to more specifically refer to explaining why input is misclassified by a neural network. The title is now: ‘ Explaining the Mistakes of Neural Networks Using Latent Sympathetic Examples’. We have also reformulated the paper to communicate this intent more clearly.  Secondly, we have included more examples on MNIST.
Thirdly, we have changed some of the examples on CelebA to ones that are more illustrative of the technique’s ability to perturb images. We have changed the normalization on the perturbations to make it more salient what is changed. We have also made a connection to heat mapping methods that exist in the prior literature.
Fourthly, the reason why epsilon must stay as small as possible is to a) reduce the reconstruction error by the generative model and b) such that it generates only those features that are absolutely necessary to perturb the image to the correct class. For instance, assume we have an image of a person that has a mustache but is misclassified as not having a mustache. If we use our method to go to the absolute minimum error, we may end up with an entirely new face projected over the original image and not just the area of the mustache. Though technically correct, this would not yield meaningful explanations, since one can always make a face more mustache like by continuing to project the prototypical mustached face on it. Therefore it is important that epsilon be encouraged to stay small. We have more clearly explained this in the paper.
Fifthly, we attempt to explain what features are often perturbed in the latent space by including an example. We have done an analysis of why certain males were misclassified as female. We found that often males with long hair were misclassified and that the hair was perturbed to get them to be correctly classified. This also reveals a novel application: discovering non-trivial dataset weaknesses, or biases. In this case our method points to insufficient males with long hair in the dataset.

Another concern that was raised was that we do not show how our method would perform against a standard adversarial attack. It is not our intent to adversarially attack, nor to guard against adversarial attacks. Normal heatmapping methods would not allow you to do so either.  The goal of our method is to locate potential areas in the input image that caused a misclassification, different from the unconstrained manner (Simonyan 2013) where the input image can be perturbed in any possible way (Ancona et al. (2017)). This would mean that the perturbations very rarely have any semantically meaningful footprint, since the perturbations would be small changes in pixel values all over the image. Because we are interested in those perturbations that could lead to reasonable misclassifications, we put constraints on it. There is no guarantee that these perturbations are meaningful, however, experimental results show us that this is often the case.

The issue of performance was raised. Specifically, the issue of local minima. To clarify  there are several causes for bad performance. 1) The generative model is not strong enough.  2) local minima. 3) Attributes that are hard to demarcate for annotators. As the size of the latent space increases the local minima problem becomes less prevalent because there are simply more ways to ‘go down’ in the error space. This is made clear by the differing success rates using a VAE with a 2-D and a 10-D latent spaces (Section 3.2.1).

Finally, the issue of the performance of the generative model is also raised. We use a VAEGAN for the CelebA data, which is more powerful than a VAE. In order to check whether the generative model is to blame, one can simply compare the reconstruction error to the perturbation and see whether the generative model was indeed capable of capturing the data well.

We hope that we have addressed all of your questions, and improved where improvement was possible. Thank you again for your feedback.

Kind regards,


Anonymous.

---

### Decision · Program_Chairs · 2018-01-29
**ICLR 2018 Conference Acceptance Decision**

**Decision:**

Reject

**Comment:**

The paper proposes a way to find why a classifier misclassified a certain instance. It tries to find pertubations in the input space to identify the appropriate reasons for the misclassification. The reviewers feel that the idea is interesting, however, it is insufficiently evaluated.  Even for the datasets they do evaluate not enough examples of success are provided. In fact, for CelebA the results are far from flattering.